

# A novel group tour trip recommender model for personalized travel systems

Mohammed Alatiyyah

Department of Computer Science, College of Computer Engineering and Sciences, Prince Sattam Bin Abdulaziz University, Al-kharj, Riyadh, Saudi Arabia

## ABSTRACT

Planning personalized travel itineraries for groups with diverse preferences is indeed challenging. This article proposes a novel group tour trip recommender model (GTTRM), which uses ant colony optimization (ACO) to optimize group satisfaction while minimizing conflicts between group members. Unlike existing models, the proposed GTTRM allows dynamic subgroup formation during the trip to handle conflicting preferences and provide tailored recommendations. Experimental results show that GTTRM significantly improves satisfaction levels for individual group members, outperforming state-of-the-art models in terms of both subgroup management and optimization efficiency.

## INTRODUCTION

The modern era has seen a rise in international travel, driven by the availability of affordable and efficient transportation systems. However, this surge in international travel presents challenges for inexperienced travelers in planning their trips (*Mao et al., 2024*; *Zhou et al., 2023*; *Yoon et al., 2012*). According to *Borràs, Moreno & Valls (2014)*, the Internet offers a vast array of information for tourists regarding travel destinations, points of interest (POIs), activities, and events. Nonetheless, this information is often unpersonalized and overwhelming, particularly when booking group tours (*Sun & Wandelt, 2021*; *Borràs, Moreno & Valls, 2014*).

Traveler recommender systems (TRSs) uniquely provide recommendations for sequences of diverse items such as POIs, meals, and accommodation (*Wu, Lyu & Liu, 2022*). The function of group recommender systems (GRS) for travelers involves managing different tourists within a group to maximize overall satisfaction and minimize conflicts arising from individual constraints and preferences (*Quijano-Sánchez et al., 2020*). Moreover, the task of generating group travel recommendations poses significant challenges for GRS, as group decision-making processes are inherently more complex (*Chen, Cheng & Chuang, 2008*; *Popescu, 2013*). Additionally, issues intrinsic to GRS further complicate this process.

The tourist trip design problem (TTDP), as defined by *Vansteenwegen & Van Oudheusden (2007)*, involves creating personalized itineraries for tourists seeking to visit multiple POIs within a limited timeframe. The TTDP assumes that tourists aim to explore

Corresponding author
Mohammed Alatiyyah,
M.alatiyyah@psau.edu.sa

various POIs while adhering to specific time constraints and considering the unique attributes of each POI, such as category, location, accessibility, and cost. Each trip has a designated maximum duration ($T_{max}$) for sightseeing. The ultimate goal of the TTDP is to maximize the overall satisfaction of tourists by recommending itineraries that include high-scoring POIs while taking into account the individual constraints and preferences of the travelers. The system not only selects the most suitable POIs but also determines the optimal routes between them, ensuring an efficient and enjoyable travel experience (*Sarkar et al., 2023*). To achieve this objective, solutions to the TTDP must adhere to the travelers' specific requirements and the characteristics of the POIs (*Gavalas et al., 2014*).

On the other hand, the group tour trip design problem (GTTDP) extends the TTDP, with the primary distinction being that the GTTDP problem addresses a group of travelers. However, it is worth mentioning that managing conflicting constraints or preferences within the group is indeed challenging (*Halder et al., 2024*). Given that specific trip activities are designed to be undertaken collectively by groups of users (*Jameson, Baldes & Kleinbauer, 2003*; *Ghazarian & Nematbakhsh, 2015*), novel challenges have arisen in the domain of GRS. Distinct factors, such as the aggregation of user profiles, users' roles, and constraints, have become increasingly important. GRS can recommend tour itineraries that align with the constraints and preferences of all group members. Furthermore, GRS must accommodate a diverse user base with potentially dissimilar preferences (*Su et al., 2020*; *Amer-Yahia et al., 2009*; *Ghazarian & Nematbakhsh, 2015*). *Ghazarian & Nematbakhsh (2015)* have identified four key challenges facing group recommender systems (GRSs): collecting user preference information, creating recommendations, explaining these recommendations, and facilitating group consensus (*Ekstrand, Riedl & Konstan, 2011*; *Ghazarian & Nematbakhsh, 2015*). *Salamó, McCarthy & Smyth (2012)* point out that recommending to a group with identical preferences is akin to recommending to a single individual, as there is no need for group-specific recommendations in such cases. However, in reality, groups often exhibit diverse preferences, leading to conflicting needs and making it challenging to satisfy all members (*Salamó, McCarthy & Smyth, 2012*; *Castro, Yera & Martínez, 2017*; *Renjith, Sreekumar & Jathavedan, 2020*).

Recent developments in GRSs have introduced more sophisticated techniques aimed at handling dynamic user preferences and enhancing group satisfaction (*Halder et al., 2024*; *Zaizi, Qassimi & Rakrak, 2023*). Studies such as *Chen et al. (2023)* and *Sharma et al. (2024)* propose models that utilize reinforcement learning and neural network algorithms to adapt recommendations in real time based on user interactions and environmental changes. These approaches reveal a trend toward more adaptable recommendation systems, where personalized interactions play a vital role. Additionally, systems such as those introduced by *Imran et al. (2023)* use predictive analytics to anticipate and mitigate preference conflicts within groups, further underscoring the need for subgroup formation strategies.

Based on the aforementioned discussion, it can be observed that the task of planning personalized group trips becomes particularly complex when group members have

conflicting preferences and constraints. In fact, current GRS models either fail to address this issue or are limited by their inability to adapt dynamically during the trip. Another major limitation of existing systems is their inability to adapt to in-trip conflicts by dynamically forming subgroups. Current models, such as those proposed by *Halder et al. (2024)* and *Vansteenwegen et al. (2009)*, either aggregate user preferences into a single itinerary or fail to account for the evolving nature of group preferences. These methods do not allow for real-time adaptations, making them unsuitable for group trips where preferences diverge over time. To address these challenges, we propose a novel group tour trip recommender model (GTTRM) that dynamically adapts to changing preferences by using ant colony optimization (ACO). Our model allows for subgroup formation during the trip, thus optimizing the satisfaction of individual group members. This flexibility is a key advancement over static models and provides a more personalized and conflict-free travel experience. The key contributions of this article are summarized as follows.

- A novel framework for dynamic subgroup formation during group trips with the consideration of various activities.
- By incorporating item constraints data model (ICDM), the proposed model reduces data dimensionality, thereby improving computational efficiency while maintaining personalization.
- Integrating a group ant colony optimization (GACO) approach that balances group satisfaction with individual preferences, dynamically adapting to conflicts as they arise.
- A comprehensive experimental evaluation that demonstrates significant improvements in satisfaction levels of group travelers.

As compared to relevant models, the proposed GTTRM model offers the following main features:

- *Time-centric approach:* Unlike traditional models that focus on POIs, the proposed GTTRM model focuses on maximizing user satisfaction over time, dynamically adjusting to traveler preferences during the trip.
- *Consideration of various activities:* The proposed GTTRM incorporates activities such as visiting POIs, traveling between POIs, and minimizing wasted time to create more comprehensive itineraries while ensuring a holistic and efficient travel plan.
- *Overcoming limitations of existing methods:* The proposed GTTRM addresses the shortcomings of current models by considering a wider range of factors that influence traveler satisfaction, such as dynamic subgroups, time management, connection preferences, and waiting times.
- *Improved personalization:* Experimental results demonstrate the GTTRM's effectiveness in generating personalized tour recommendations that align with user preferences, outperforming existing models in terms of both effectiveness and intuitiveness.

The remaining of the article is structured as follows. Related studies on GRSs and the TTDP are reviewed in "Related Work", focusing on aggregation methods and optimization

approaches. The proposed model is presented in "The Proposed Group Tour Trip Recommender Model", including its architecture, algorithms, and constraints. The experimental results are presented in "Experimental Results". This section analyzes the experimental findings, comparing the performance of different aggregation methods and interprets the obtained results and their implications. Finally, the the key findings, contributions, and future research directions are concluded in the "Conclusion".

**Remark 1.** While our previously published study focused on individual tour trips (*Alatiyyah, 2024*), this article mainly addresses tour trips organized for groups of travelers. Typically, people travel in groups, such as friends, couples, or families; therefore, when recommendations are required for multiple individuals, GRSs become particularly relevant and beneficial. However, planning personalized travel itineraries for groups of individuals with diverse preferences and constraints in indeed challenging.

## RELATED WORK

The landscape of GRS research has recently expanded with models focused on adaptive and real-time group decision-making, incorporating a variety of machine learning and optimization techniques. For example, *Gulzar et al. (2023)* introduced a clustering-based GRS that groups users based on behavioral patterns, optimizing recommendations through cluster analysis. While effective, this approach lacks the real-time adaptability offered by the GTTRM, which forms subgroups dynamically during the trip rather than pre-grouping users based on static data. Another recent contribution by *Wei et al. (2023)* integrates attention mechanisms within GRS to prioritize influential users' preferences in group decisions. Although this method improves satisfaction by weighting preferences, it does not allow for dynamic regrouping during the trip, which limits its adaptability in scenarios with conflicting user preferences. Our GTTRM addresses these limitations by forming and dissolving subgroups as needed, enabling a flexible response to changing preferences. Furthermore, *Wang et al. (2024)* developed a relationship-driven GRS model that enhances group cohesion in recommendations for football formation. Although their model effectively considers relationship strength, it does not provide a mechanism for subgroup formation, making it less suitable for diverse travelers groups with highly varied interests. Finally, studies by *Migliorini et al. (2024)* and *Achmad et al. (2023)* have incorporated real-time context adaptation, where recommender systems adjust based on location or environmental factors. While these models provide valuable insights for real-time travel recommendations, they lack the subgroup formation capability that is central to our GTTRM. Our model not only adapts to real-time conditions but also allows for subgroup configuration, which is crucial in addressing conflicts that arise from diverging preferences among group members.

In this study, we adopt a classification that divides GRS approaches into two primary categories: (1) Aggregation approaches that focus on refining or modifying existing algorithms to enhance the quality of recommendations for groups of users within the GRS framework. (2) Optimization approaches that employ optimization techniques to address the GTTDP directly, aiming to optimize the overall satisfaction of group members while considering their diverse preferences and constraints. By categorizing GRS approaches in

**Table 1  Common aggregation techniques are used in GRSs.**

| Strategy | Description |
|---|---|
| Plurality voting | The item with the highest number of votes is chosen. |
| Average | The average of individual ratings is calculated. |
| Multiplicative | The product of individual ratings is calculated. |
| Borda count | Each traveler's preferences are ordered, and a value is assigned to every particular item. The lowest-rated receives 0, the next receives 1, and so on. |
| Copeland rule | The number of times an item beats other items is counted, minus the number of times it loses. |
| Approval voting | The number of times an item has been rated is counted. |
| Least misery | The lowest rating among all travelers is selected. |
| Most pleasure | The highest rating among all travelers is selected. |
| Average without misery | The average of traveler ratings is calculated, excluding any ratings under a specific threshold. |
| Fairness | The top-rated items are chosen from all travelers. |
| Most respected person | The most-respected rating from a particular traveler is used. |

this manner, we provide a clear and comprehensive framework for understanding and evaluating the various techniques employed in this field.

The primary challenge for GRSs lies in developing a recommendation system that adequately satisfies all group members. Various methodologies have been employed to address this challenge. As per *Zaizi, Qassimi & Rakrak (2023)*, *Jameson & Smyth (2007)*, GRSs can be categorized into three primary aggregation methods: (1) aggregating individual preferences, (2) aggregating individual ratings, and (3) aggregating individual recommendations. The first approach involves merging the preferences of all group members into a unified group preference, often referred to as constructing group preference models (*Jameson & Smyth, 2007*). A crucial challenge for GRSs is aligning individual user preferences with the collective group preferences (*Masthoff, 2011*). In the first approach, individual preferences are aggregated into a group model (G), which is subsequently used to predict ratings for each candidate item. The second approach, individual rating aggregation, involves combining the ratings for each item from every group member. The third approach generates recommendations for each group member individually and then aggregates these recommendations to form the group's recommendations.

Prior to delving into the aggregation approaches, it is essential to explore the various strategies they encompass. For example, *Masthoff (2011)* identifies eleven aggregation techniques commonly used in GRSs, each applied to distinct types of aggregations (*Chen et al., 2022*). Table 1 outlines these strategies and their respective functionalities. The majority of previous studies employ one of these strategies, often with minor modifications (*Masthoff, 2011*).

## Recommendation aggregation

Aggregation of individual recommendations involves combining personalized recommendation lists into group recommendations using various approaches. *Christensen & Schiaffino (2011)* employed six distinct aggregation strategies for GRSs (*Baltrunas, Makcinskas & Ricci, 2010*). One such technique merges individual recommendations (*Halder et al., 2024*; *Christensen & Schiaffino, 2011*). This approach is based on generating recommendations for individuals and is straightforward to implement as it extends existing recommender systems. Additionally, *Baltrunas, Makcinskas & Ricci (2010)* describe four distinct rank aggregation approaches: (i) Spearman footrule, which minimizes the average distance among individual items, (ii) Least misery, (iii) Average, and (iv) Borda count. Regarding the most effective technique, *Masthoff (2011)* conducted experiments to identify the optimal strategy. The research revealed that users prioritized increasing fairness and reducing misery. However, Masthoff noted that the multiplicative strategy yielded the highest levels of user satisfaction.

## Preference aggregation

In *Sarkar et al. (2023)*, *Garcia, Linaza & Arbelaitz (2012)*, *Baskin & Krishnamurthi (2009)*, *Garcia, Sebastia & Onaindia (2011)*, *Garcia et al. (2009)*, *Christensen & Schiaffino (2011)*, *Ding et al. (2024)*, *Yu et al. (2006)*, *Salamó, McCarthy & Smyth (2012)*, *Kim et al. (2010)*, *McCarthy & Anagnost (1998)*, *Boratto & Carta (2015)*, *Shang et al. (2014)*, aggregation strategies are employed to construct a common profile by consolidating individual user preferences. Specifically, each user has particular preferences, such as parks, museums, or Indian cuisine. This methodology aggregates these individual preferences to formulate a unified preference for each group. Travelers may select more than one preference, and some preferences may overlap among users. Then, the aggregation function synthesizes the preference for the particular group.

*Yu et al. (2006)* propose a method for generating preferences for a particular group by evaluating users' references for TV viewing. This method minimizes total distance to compute the dissimilarity among users' preferences. Each preference a user likes is assigned a value of 1, dislikes are assigned −1, and unknown preferences are assigned 0. *Kim et al. (2010)* propose a GRS that improves group recommendation effectiveness and member satisfaction. Their system uses a collaborative filtering (CF) model to aggregate group member preferences, creating a candidate recommendation set. If a group member has interacted with an item, like reading a book, it is assigned a value of 1. The group rating for each item is then calculated, and the nearest-neighbor algorithm determines the similarity between group profiles.

## Rating aggregation

In *Christensen & Schiaffino (2011)*, *Popescu (2013)*, *Amer-Yahia et al. (2009)*, *Berkovsky & Freyne (2010)*, *Naamani-Dery et al. (2010)*, *O'Hara et al. (2004)*, *Sprague, Wu & Tory, 2008*, *Kim & Saddik (2015)*, *Gartrell et al. (2010)*, the average and least misery are identified as the predominant aggregation functions proposed for recommendations or rating aggregations (*Amer-Yahia et al., 2009*). Nonetheless, *Amer-Yahia et al. (2009)* introduces a

consensus function, composed of relevance and disagreement functions. The relevance function consolidates the ratings of all group travelers using both the average and least misery strategies, while the disagreement function assesses the extent to which the group members have collectively liked or disliked an item. Essentially, the consensus function evaluates the suitability of an item for group recommendation by determining its relevance and the level of disagreement among group members. *Berkovsky & Freyne (2010)* suggest a recommender model for food that seeks to identify the most appropriate data aggregation strategy for a family group. They apply four strategies: two static and two interaction-based. The first is a uniform model assigning equal weight to each user. The second is a heuristic model based on role, with weights assigned as follows: applicant = 0.5, partner = 0.3, and child = 0.1. The third strategy is a role-based model which identified a particular weight for each user based on their activity, represented by the number of observed ratings from that user. In *Gartrell et al. (2010)* propose a GRS model that analyzes various group characteristics. Such a model developed to anticipate group preferences through the implementation of a group-consensus function, utilizing association rules to uncover significant patterns among users within the dataset.

## Optimization of GRSs: a comparative analysis to existing studies

In recent years, the landscape of group travel recommender systems has seen a variety of approaches aimed at enhancing the experience of travelers. However, existing models, including the Multi-Constraint Multiple Team Orienteering Problem with Time Windows (MCMTOPTW), have predominantly centered around static itineraries, failing to account for the fluid dynamics of user preferences that often evolve during a trip.

To the best of the author's knowledge and believe, only a limited number of studies (*Renjith, Sreekumar & Jathavedan, 2020*; *Anagnostopoulos et al., 2017*; *Sylejmani, Dorn & Musliu, 2017*) have addressed GRSs using optimization approaches. These studies have predominantly developed their models based on the orienteering problem (OP). Given that the challenges associated with individual tourist recommendations remain partially unresolved, some researchers have extended their models to accommodate groups of travelers.

Addressing GRSs as an optimization problem proves to be an effective method due to two primary reasons: (1) the GRSs for travelers encompass the challenges of TRSs, which are fundamentally data-driven issues, and (2) personalization is essential to cater to the diverse preferences of users. The OP is the most analogous model for solving the time-dependent traveling salesman problem (TTDP), as it represents a specific case of TTDP predicated on fundamental constraints.

While previous studies (*Renjith, Sreekumar & Jathavedan, 2020*; *Anagnostopoulos et al., 2017*; *Sylejmani, Dorn & Musliu, 2017*) have primarily focused on optimization approaches based on the OP, they fall short in addressing the complexities and variabilities inherent in group travel scenarios. Specifically, these models often overlook critical factors such as individual user constraints, multi-values for POIs, connection values, waiting times, and the aggregation of preferences. The proposed GTTRM is meticulously designed to remedy

these shortcomings, providing a framework that adapts to diverse user needs and incorporates multiple relevant factors in group travel optimization.

Unlike existing models such as the MCMTOPTW and others that primarily optimize static group itineraries, the proposed GTTRM introduces a novel approach by allowing for dynamic subgroup formation. This means that during the trip, subgroups can form and dissolve based on evolving user preferences, which is not addressed in current models. The employment of the GACO algorithm within the framework of GTTRM facilitates real-time adaptations, greatly enhancing overall group satisfaction. No existing models in the current literature, to the best of our knowledge, provide such dynamic subgroup flexibility combined with GACO-based optimization.

## THE PROPOSED GROUP TOUR TRIP RECOMMENDER MODEL

This section illustrates the proposed GTTRM, which is a model that can solve the problem of the GTTDP (*Alatiyyah, 2019*). In fact, the GTTRM offers two aggregation methods: (1) Group aggregation (GA) and (2) User aggregation (UA). The GA method aggregates all user constraints and preferences into a single group profile. Conversely, the UA method involves the algorithm in the construction of recommended tours, where the GTTRM algorithm strategically groups users to maximize satisfaction and minimize conflicts among group members. Figure 1 illustrates the GA method, while Fig. 2 depicts the UA method.

In this study, the principal approach involves dividing the group into subgroups for portions of the trip duration to maximize satisfaction levels when complex conflicts arise. The proposed recommendation model represents a novel approach that determines the feasibility of partitioning the overall group into particular subgroups to optimize the level of satisfactions among group members by assessing their similarities. Furthermore, the proposed model determines the optimal locations and timings for dividing the overall group.

Table 2 delineates the primary distinctions between the aggregation methods utilized in the GTTRM. Firstly, the GA method does not facilitate the creation of sub-routes for the group, as its primary objective is to aggregate all group members into a single profile. Conversely, the UA method aims to optimize the satisfaction level of each individual in the group by considering alternative options, such as the creation of sub-routes. Secondly, the GA method generates a single profile for the group based on one of the aggregation techniques (refer to Table 1), whereas the UA method treats each user as an individual, taking into account the preferences of other group members when calculating probabilities. Thirdly, the GA method minimizes the search space by consolidating all users' preferences and constraints into a unified profile. Lastly, the UA method yields superior outcomes by maximizing the satisfaction levels, as it recommends items tailored to individual preferences of every particular traveler.

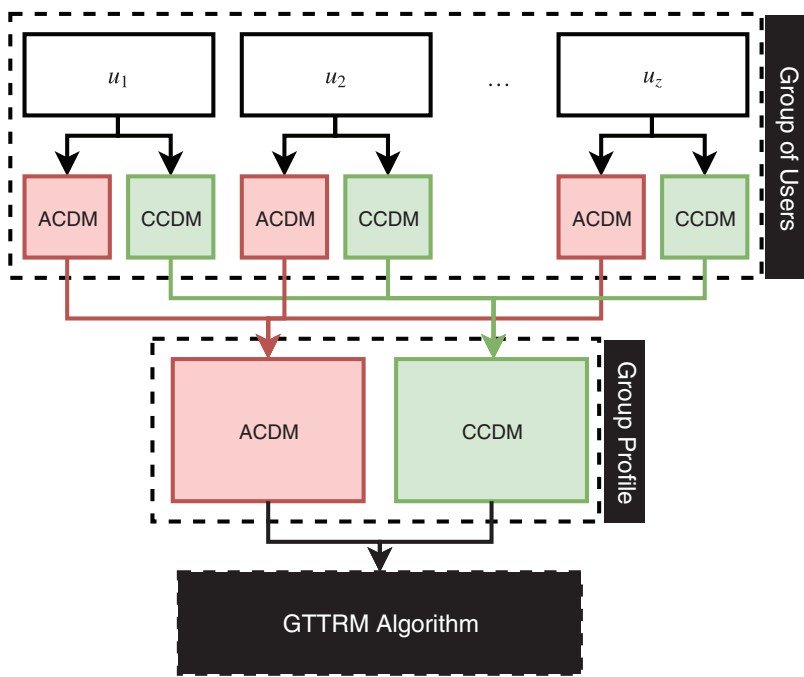

**Figure 1 Illustration of group aggregation linked to the proposed algorithm.**

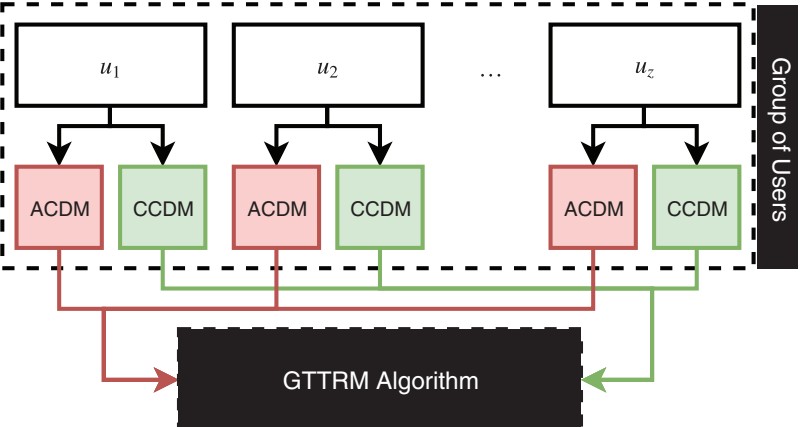

**Figure 2 Illustration of user aggregation linked to the proposed algorithm.**

**Table 2 Comparison among group/user aggregation approaches.**

| Features | Group aggregation | User aggregation |
|---|---|---|
| Dividing the group | | • |
| Ignoring taste differences | • | |
| Reduced search space | • | |
| Maximizing satisfaction | | • |

## Constraints data model

Building upon the ICDM developed in our previous work (*Alatiyyah, 2024*) to handle various constraints proposed by individual users, a constraints data model (CDM) is consequently developed in this article to handle the preferences and connection constraints from a particular group of travelers. The implemented constraints model contains two primary components:

- *Activity constraints data model (ACDM):* The ACDM focuses on aligning activity data with user constraints. It is essentially an extension of the ICDM to accommodate the needs of a group of users.
- *Connection constraints data model (CCDM):* The CCDM is responsible for validating connection data against user constraints. It ensures that the connections between activities or POIs adhere to the specified constraints and preferences of group members.

The key differences between the CCDM and ACDM lie in the specific data and connection constraints they handle. The ACDM primarily focuses on aligning activity data with user constraints, while the CCDM validates connection data against user constraints. This distinction is crucial for effectively managing both individual and group-level constraints within the context of group travel.

The CDM is defined as follows: $u_z \in G$ denotes a traveler within the group $G$ where $z = 1, 2, \ldots, |G|$, and each traveler $u_z$ may have $n$ constraints (soft constraint (SC), hard constraint (HC)). $ActHC_{u_z}$ and $ConHC_{u_z}$ denote sets of HCs for activity and connection from user $u_z$, respectively, where $hc(Act)_{ptim}^{u_z} \in ActHC_{u_z}$ and $hc(Con)_{ptim}^{u_z} \in ConHC_{u_z}$; $\forall u_z \in U; \forall p \in P; \forall t \in p; \forall i \in I$, with $m = 1, 2, \ldots, (|ActHC_{u_z}|$ or $|ConHC_{u_z}|)$. Similarly, $ActSC_{u_z}$ and $ConSC_{u_z}$ are sets of SCs for Activity and Connection from user $u_z$, respectively, where $sc(Act)_{ptim}^{u_z} \in ActSC_{u_z}$ and $sc(Con)_{ptim}^{u_z} \in ConSC_{u_z}$, with $v = 1, 2, \ldots, (|ActSC_{u_z}|$ and $|ConSC_{u_z}|)$.

$$ActHC_{pti}^{u_z} = \prod_{m=1}^{|ActHC_{u_z}|} hc(Act)_{ptim}^{u_z} \tag{1}$$

$$ActHC_{pti}^{G} = \prod_{z=1}^{|G|} ActHC_{pti}^{u_z}. \tag{2}$$

Equation (1) consolidates all the HCs for the individual user $u_z$, while Eq. (2) aggregates all the HCs for the entire user group. Specifically, Eq. (2) denotes the process of group aggregation (GA).

$$ConHC_{ptij}^{u_z} = \prod_{m=1}^{|ConHC_{u_z}|} hc(Con)_{ptijm}^{u_z} \tag{3}$$

$$ConHC_{ptij}^{G} = \prod_{z=1}^{|G|} ConHC_{ptij}^{u_z}. \tag{4}$$

Equations (3) and (4) are utilized to compute the HC for both individual users and groups.

$ActSC_{pti}^{u_z}$ and $ConSC_{ptij}^{u_z}=$ Aggregation methods for the user $(u_z)$       (5)

$ActSC_{pti}^{G}$ and $ConSC_{ptij}^{G}=$ Aggregation methods for the group $(G)$.       (6)

Equations (5) and (6) denote the aggregations of all soft constraints for group $G$ and traveler $u_z$.

$$\sum_{v=1}^{|ActSC_{u_z}|} W_v = 1 \text{ and } \sum_{v=1}^{|ConSC_{u_z}|} W_v = 1 \tag{7}$$

$$\sum_{u_z=1}^{|G|} W_{u_z}^{G} = 1 \text{ and } \sum_{u_z=1}^{|G|} W_{u_z}^{G} = 1. \tag{8}$$

Equation (7) illustrates that each SC may possess a distinct weight, signifying the relative importance of one SC in comparison to another. Equation (8) demonstrates the capacity to adjust for the varying expertise levels among group travelers, whereby a traveler with a higher expertise is accorded a greater weight.

$$A_{pti}^{u_z} = ActHC_{pti}^{u_z} \times ActSC_{pti}^{u_z} \tag{9}$$

$$A_{pti} = ActSC_{pti}^{G} \times ActSC_{pti}^{G} \tag{10}$$

$$C_{ptij}^{u_z} = ConHC_{ptij}^{u_z} \times ConSC_{ptij}^{u_z} \tag{11}$$

$$C_{ptij} = ConHC_{ptij}^{G} \times ConSC_{ptij}^{G}. \tag{12}$$

Equation (9) highlights the aggregation among Eqs. (1) and (5) to determine the weight that implies the satisfaction level for traveler $u_z$ in each item $i$ on day $p$ at time $t$ based on the traveler's constraints. Equation (10) collects the soft and hard constraints of the group.

Equation (9) aggregates Eqs. (1) and (5) to determine the weight representing the satisfaction level for traveler $u_z$ in each item $i$ on day $p$ at time $t$. This weight is determined based on the user's individual HCs and SCs. Equation (10) subsequently combines the SCs and HCs of all group members to calculate the overall group weight for each item. This group weight reflects the collective satisfaction level of the group members regarding the item.

$$ActHC_{pti}^{u_z}, ActHC_{pti}^{G}, ConHC_{pti}^{G}, ConHC_{pti}^{u_z} \in \{0,1\} \tag{13}$$

$$ActSC_{pti}^{u_z}, ActSC_{pti}^{G}, ConSC_{pti}^{G}, ConSC_{pti}^{u_z} \in \mathbb{Q}; \tag{14}$$

Where is $0 \leq ActSC_{pti}^{u_z}, ActSC_{pti}^{G} \leq 1$;

Where is $0 \leq ConSC_{pti}^{G}, ConSC_{pti}^{u_z} \leq 1$.

The principal functionality of partitioning the group into subgroups is facilitated by our proposed algorithm. The details of the algorithm are delineated in "The Proposed Algorithm" (Tables 3 and 4).

**Table 3 Soft constraints aggregation methods.**

| Method | Interpretation | Associated equation |
|---|---|---|
| Sum | Identify the sum of all soft constraints | $\sum_{v=1}^{\lvert ActSC_{u_z} \rvert} W_v \times sc(Act)_{ptim}^{u_z}$ |
| Least misery | Minimum value of soft constraints | $Min(ActSC_{u_z})$ |
| Most pleasure | Maximum value of soft constraints | $Max(ActSC_{u_z})$ |
| Multiplicative | Multiplies each soft constraints value | $\prod_{v=1}^{\lvert ActSC_{u_z} \rvert} W_v \times sc(Act)_{ptim}^{u_z}$ |

**Table 4 Notations of the proposed GTTRM.**

| Notation | Meaning |
|---|---|
| $X_{pti}^{u_z}$ | A binary decision variable indicating whether traveler $u_z$ visits POI $i$ at time $t$ on day $p$. |
| $A_{pti}^{u_z}$ | The satisfaction level of traveler $u_z$ for visiting the POI $i$ at time $t$ on day $p$ (see Eq. (9)) |
| $ActG_{pti}^{u_z u_o}$ | A binary decision variable indicating whether traveler $u_z$ and traveler $u_o$ visit POI $i$ at time $t$ on day $p$ |
| $V_{u_o}^{u_z}$ | The satisfaction level of traveler $u_z$ when traveling with traveler $u_o$ |
| $Y_{ptij}^{u_z}$ | A binary decision variable indicating whether traveler $u_z$ travels from POI $i$ to POI $j$ at time $t$ on day $p$ |
| $C_{ptij}^{u_z}$ | The satisfaction level of traveler $u_z$ to travel from POI $i$ to POI $j$ at time $t$ on day $p$ (see Eq. (11)). |
| $ConG_{ptij}^{u_z u_o}$ | A binary decision variable indicating whether traveler $u_z$ and traveler $u_o$ move together from POI $i$ to POI $j$ at time $t$ on day $p$. |
| $T_{u_o}^{u_z}$ | The satisfaction level of traveler $u_z$ when traveling with traveler $u_o$ |
| $Z_{pt}^{u_z}$ | A binary decision variable indicating whether traveler $u_z$ has waiting time at time $t$ on day $p$. |
| $W_{pt}^{u_z}$ | The satisfaction level of traveler $u_z$ for doing nothing at time $t$ on day $p$. |

## Mathematical model

The proposed model can be formally identified as follows. Let $G = (I, TT)$ represent a directed weighted graph, where $i \in I$ and $i = 1, \dots, \lvert I \rvert$ denote the nodes corresponding to POIs within a particular city. The travel time among two nodes $i$ and $j$ is represented by $TT_{ij}$, while $ST_i$ indicates the time allocated at node $i$. Given an initial node $s$ and an ending node $t$, we define $s = 1$ and $t = \lvert I \rvert$. The duration of the tour may span one or more days, hence let $p \in P$; $p = \{1, \dots, \lvert P \rvert\}$ denote the set of tour days. Additionally, each tour day includes a set of time instances $t \in p$; $t = 1, \dots, \lvert p \rvert$, representing specific moments within day $p$. The daily time constraint for the trip is denoted by $T_{max}$.

The subsequent equations delineate the GTTRM along with its constraints. Equation (15) specifies the objective function, which aims to maximize the overall score derived from three distinct actions: (i) Activity, (ii) Connection, and (iii) Waiting. This equation encompasses three principal functions: $f_1(a)$, $f_2(c)$, and $f_3(w)$, each one is corresponded to the aforementioned actions, respectively.

$$Max(f_1(a) + f_2(c) + f_3(w)). \tag{15}$$

The three functions $f_1(a)$, $f_2(c)$, and $f_3(w)$ are shown in Eqs. (16), (18), and (20).

$$f_1(a) = \sum_{p=1}^{|P|} \sum_{t=1}^{|p|} \sum_{i=1}^{|I|} \sum_{z=1}^{|G|} X_{pti}^{u_z} \times A_{pti}^{u_z} + \sum_{o=1}^{|G|} ActG_{pti}^{u_z u_o} \times V_{u_o}^{u_z}. \tag{16}$$

Equation (16) denotes the satisfaction of a particular group $G$ by traveling to some POIs. Additionally, if $u_z$ travels to POIs with a particular froup member, the level of their satisfaction levels may enhance.

$$ActG_{pti}^{u_z u_o} = X_{pti}^{u_z} = X_{pti}^{u_o} \tag{17}$$
$$\forall t = 1, \ldots, |p|; \forall p = 1, \ldots, |P|; \forall z = 1, \ldots, |G|; \forall o = 1, \ldots, |G|; \forall i = 1, \ldots, |I|.$$

Equation (17) guarantees that if traveler $u_z$ visits a particular POI at least one member of the group would also visit the same POI.

$$f_2(c) = \sum_{p=1}^{|P|} \sum_{t=1}^{|p|} \sum_{i=1}^{|I|} \sum_{j=1}^{|I|} \sum_{z=1}^{|G|} Y_{ptij}^{u_z} \times C_{ptijr}^{u_z} + \sum_{o=1}^{|G|} ConG_{ptij}^{u_z u_o} \times T_{u_o}^{u_z}. \tag{18}$$

Equation (18) calculates the overall group satisfaction level by considering the most preferred connections among POIs $i$ and $j$. This equation considers the individual satisfaction levels of group members for traveling together and the overall preference for specific connections. Additionally, if traveler $u_z$ moves from one POI to another with a group member, their satisfaction may also enhance.

$$ConG_{ptij}^{u_z u_o} = Y_{ptij}^{u_z} = Y_{ptij}^{u_o} \tag{19}$$
$$\forall t = 1, \ldots, |p|; \forall p = 1, \ldots, |P|; \forall z = 1, \ldots, |G|;$$
$$\forall o = 1, \ldots, |G|; \forall i = 1, \ldots, |I|; \forall j = 1, \ldots, |I|.$$

Equation (19) guarantees that if traveler $u_z$ visits a POI, at least one member of the group would also visit this particular POI.

$$f_3(w) = \sum_{p=1}^{|P|} \sum_{t=1}^{|p|} \sum_{z=1}^{|G|} Z_{pt}^{u_z} \times W_{pt}^{u_z}. \tag{20}$$

Equation (20) find the overall waiting time for all members in the group to identify the overall satisfaction of the group members.

$$\sum_{i=1}^{|I|} X_{pti}^{u_z} + \left( \sum_{i=1}^{|I|} \sum_{j=1}^{|I|} Y_{ptij}^{u_z} \right) + Z_{pt}^{u_z} = 1 \tag{21}$$
$$\forall t = 1, \ldots, |p|; \forall p = 1, \ldots, |P|; \forall z = 1, \ldots, |G|$$

$$\sum_{j=1}^{|I|} Y_{p11j}^{u_z} = 1 \tag{22}$$
$$\forall p = 1, \ldots, |P|; \forall z = 1, \ldots, |G|$$

$$\sum_{i=1}^{|I|-1} \left( \frac{\sum_{t=1}^{|p|} Y_{pti|I|}^{u_z}}{TT_{i|I|}} \right) = 1 \tag{23}$$

$$\forall p = 1, \ldots, |P|; \forall z = 1, \ldots, |G|.$$

Equation (23) is a crucial constraint that ensures that each user can only perform one activity at a time. This constraint prevents conflicts and ensures efficient itinerary planning. Furthermore, Eqs. (22) and (23) enforce the starting and ending points of the trip for each day $p$. Equation (22) guarantees that the trip begins at the designated starting point $s$, while Eq. (23) ensures that the trip concludes at the designated end point $e$. These constraints are essential for maintaining the overall structure and feasibility of the itinerary.

$$\frac{\sum_{t=n}^{t_1=n+TT_{sr}} Y_{ptsr}^{u_z} + \sum_{t=t_1+1}^{t_2=t1+ST_r} X_{ptr}^{u_z}}{TT_{sr} + ST_r} = \frac{\sum_{t=t_2+1}^{t_3=t_2+TT_{rm}} Y_{ptrm}^{u_z}}{TT_{rm}} \leq 1 \tag{24}$$

$$\forall n \in \{1, \ldots, |p-3|\}; \forall p = 1, \ldots, |P|; \forall s, m = 1, \ldots, |I|; \forall r = 2, \ldots, |I-1|;$$
$$\forall z = 1, \ldots, |G|.$$

Equation (24) is a constraint that guarantees the tour is connected, and guarantees the connection and visiting times are equivalent to $TT_{i,j}$ and $ST_i$, respectively.

## The proposed algorithm

Based on the individual traveler preferences and associated constraints, the proposed GTTR model is capable of strategically dividing the group into subgroups. The proposed algorithm employs ant colony optimization (ACO) to facilitate this subgrouping process for travelers with potentially conflicting preferences and constraints. This algorithm effectively addresses intra-group conflicts by generating tailored sub-routes for specific group members, ultimately enhancing the satisfaction level of each individual.

### Group ant colony optimization

The GACO algorithm is devised in this article to address the group traveling tourist problem. The primary parameters utilized in GACO are presented in Table 5. The initial values for such parameters are also shown, which were determined through a series of experiments aimed at identifying the optimal parameter values while maintaining an acceptable runtime.

The primary equations employed in the GACO algorithm are presented in this section. The parameter $\eta$ is derived from the combined influence of the rate-of-activity score and the connection score relative to distance. This parameter captures the overall attractiveness of a particular POI or route based on its proximity and the potential for engaging activities.

$$\eta_{iju_z} = \left( \frac{A_i^{u_z}}{TT_{ij}} \right) + \left( \frac{C_{ij}^{u_z}}{TT_{ij}} \right). \tag{25}$$

**Table 5 Initial parameters for GACO algorithm.**

| Parameter | Initialization | Meaning |
|---|---|---|
| $\alpha$ | 0.5 | Importance of $\tau_{iju_z}$ |
| $\beta$ | 2 | Importance of $\eta_{iju_z}$ |
| $\gamma$ | 2 | Importance of $\lambda_{iju_z}$ |
| $\eta_{iju_z}$ | | Rate of score to distance |
| $\tau_{iju_z}$ | | Pheromones level from $i$ to $j$ |
| $\lambda_{u_z u_o}$ | | Social relationship among users $u$ and $o$ |
| $\rho$ | 0.5 | Value of pheromone evaporation |
| $\delta_{iju_z}$ | | Maximum of total path scores $i$ to $j$ |
| $Ant\_No$ | 20 | Number of ants |
| $Iterations$ | 10 | Number of iteration |

**Table 6 The employed functions in GACO algorithm.**

| Function | Description |
|---|---|
| $initialization()$ | Values initialization |
| $FindLastNodeTour(list)$ | Obtain the last node in the trip |
| $FindCandidateNodes(list)$ | Obtain a list of nodes that is capable on visiting it subject to the imposed constraints |
| $SelectedNode(list)$ | A node is chosen from a list |
| $FindUsers(node)$ | Get all travelers capable of visiting a particular node |
| $FOUVSN(node)$ | Obtain other travelers visiting a similar particular node |
| $FOUTSNTAN(node)$ | Get other travelers going from a particular node to another one |
| $FindAllNodeAvailable(time)$ | All nodes available at particular time is returned |

The probability of each node for each traveler in the group is calculated as

$$P_{iju_z} = \frac{\left(\tau_{iju_z}\right)^{\alpha} \left(\eta_{iju_z}\right)^{\beta} \left(\sum_{v=1}^{|G|} \lambda_{iju_z u_o}\right)^{\gamma}}{\left(\sum_{i,j=1}^{|I|} \tau_{iju_z}\right)^{\alpha} \left(\sum_{i,j=1}^{|I|} \eta_{iju_z}\right)^{\beta} \left(\sum_{i,j=1}^{|I|} \sum_{o=1}^{|G|} \lambda_{iju_z u_o}\right)^{\gamma}}. \tag{26}$$

Equation (27) and (28) show the local update pheromones.

$$\delta_{iju_z} = Max(\delta_{iju_z}, Ant_k(iju_z)) \tag{27}$$

$$\tau_{iju_z} = (1 - \rho) \times \tau_{iju_z} + \delta_{iju_z}. \tag{28}$$

The global update pheromones are then calculated as

$$\tau_{iju_z} = \rho \times \tau_{iju_z} + (1 - \rho) \times \delta_{iju_z}. \tag{29}$$

Table 6 provides a comprehensive list of the functions employed in the GACO algorithm, along with their respective tasks. The proposed GACO algorithm is illustrated in Algorithms 1–4.

---

**Algorithm 1** An overview of GACO.

```
1    initialization();
2    while day < TripLength do
3        while i < Iterations do
4            while AntK < AntNo do
5                MultiDaysRoute= Route(AntK);
6                while user < GroupSize do
7                    CurrentScore(user) = ScoreCalculate(MultiDaysRoute(user));
8                    if CurrentScore(user) > BestScore(user) then
9                        BestScore(user) = CurrentScore(user);
                    end
                end
10               LocalUpdatePheromones();
            end
11           GlobalUpdatePheromones();
        end
    end
```

---

**Algorithm 2** An overview of route function.

```
1    while true do
2        LastNodeRoutes = FindLastNodeTour();
3        while user < GroupSize do
4            FinishGroupTour * = FinisthTour(user);
        end
5        if FinishGroupTour then
6            break;
        end
7        while user < GroupSize do
8            Probability(user) = CalculateProbability(user);
9            CandidateNodesList(user) = FindCandidateNodes(user);
        end
10       SelectNodeUser[GroupSize] = 0;
11       while user < GroupSize do
12           SelectNodeUser[user] = SelectedNode(CandidateNodesList(user));
13           if CandidateNodesList.Users(SelectNodeUser[user]) > 1 then
14               AllUsersInCandidateNodesList =
                     FindUsers(CandidateNodesList(SelectNodeUser[user]));
```

---

**Algorithm 2** (continued)

| | |
|---|---|
| 15 | SelectNodeUser[AllUsersInCandidateNodesList] = 1; |
| | **end** |
| | **end** |
| 16 | **while** *user < GroupSize* **do** |
| 17 | **if** *sum(Probability(user)) == 0* **then** |
| 18 | Routes(user) = [Routes(user),EndNode]; |
| 19 | FinisthTour(user) = 1; |
| | **else** |
| 20 | Routes(user) = [Routes(user),SelectNodeUser(user)]; |
| | **end** |
| | **end** |
| | **end** |

---

**Algorithm 3** An overview of ScoreCalculate function.

1    **while** *user < GroupSize* **do**
2      Routes(user) = MultiDaysRoute(user);
3     **while** *index < length(Routes(user)) −1* **do**
4       ActivityScores(user) += Score(Routes(user)(index)) * VisitingTime(Routes(user)(index));
5       ActivityScores(user) += FOUVSN(Routes(user)(index));
6      ConnectionsScores(user) += Connections(Routes(user)(index),Routes(user)(index + 1)) * Distance(Routes(user)(index),Routes(user)(index + 1));
7       ConnectionsScores(user) += FOUTSNTAN(Routes(user)(index));
     **end**
8     WaitingScores(user) += TripTotalTime − $T_{\max}$ * WaitingTimeWeight;
   **end**

---

**Algorithm 4** An overview of ScoreCalculate function.

1    **while** *user < GroupSize* **do**
2      LastNodeRoutes = FindLastNodeTour(user);
3      AccessibleNodes(user) = FindAllNodeAvailable(time);
4      **while** *i < length(AccessibleNodes(user))* **do**
5       Probability.Node(user,i) = AccessibleNodes(user)(i);
6       Probability.User(user,i) = user;
7      Probability.Percentage(user,i) = power(Eta(LastNodeRoutes,i,user),Alpha) * power(Tau(LastNodeRoutes,i,user),Beta) * power(SocialRelationship(user,user),Gamma);
      **end**
    **end**

Algorithm 1 outlines the primary algorithm, which begins with the initialization function that assigns initial values to all relevant parameters. Subsequently, the algorithm enters into nested loops for *AntNo*, *Iterations*, and *TripLength*, iterating through each trip day, iteration, and ant, respectively. The *Route(AntK)* function, detailed in Algorithm 2, determines the route for each ant, while the *ScoreCalculate()* function, depicted in Algorithm 3, evaluates the score of each generated route.

## Comparison with existing models

While many existing models, such as multiobjective optimization problem (MOOP) and MCTOPMTW, focus primarily on optimizing the selection of POIs based on specific dimensions (*e.g.*, total scores, distance), the proposed model takes a broader approach. The proposed GTTRM considers a wider range of factors—such as multi-value (MV) connections, waiting times, and personalized travel constraints—that are often overlooked in existing techniques. Existing models are often limited by their ability to effectively handle the diverse constraints of group members. However, the proposed GTTRM addresses these limitations by incorporating a comprehensive set of features. Thus, comparing these models directly might be misleading as they address different aspects of travel planning.

Despite the differences in factors studied, our proposed GTTRM model offers several distinct advantages over existing approaches:

- *Dynamic subgroups formation:* Unlike existing models such as MCMTOPTW and others that primarily optimize static group itineraries, the proposed GTTRM introduces a novel approach by allowing for dynamic subgroup formation. This means that during the trip, subgroups can form and dissolve based on evolving user preferences, which is not addressed in current models.

- *Multi-value (MV) consideration for nodes and connections:* Unlike models like MOOP, which evaluate POIs based solely on scores or distances, GTTRM incorporates multiple attributes for each POI and the transitions between them (*e.g.*, time, cost, journey length). This multi-dimensional evaluation provides a more comprehensive understanding of traveler satisfaction.

- *Aggregation for traveler satisfaction:* While MOOP focuses on maximizing benefits within POI categories, GTTRM aggregates multiple values into a single metric representing the traveler's overall satisfaction, factoring in elements such as time spent at POIs, journey length, and costs.

- *Personalization and waiting time consideration:* One key limitation of models like MOOP is the lack of support for personalization, such as waiting time, which is crucial for travelers with reservations (*e.g.*, flights, hotels). GTTRM addresses this by including waiting time as an essential factor, enhancing its suitability for real-world travel scenarios.

- *Handling multiple decision-making parameters:* Unlike the more limited scope of MOOP and MCTOPMTW, GTTRM covers a comprehensive range of tourist decision-making parameters by categorizing them into: Activities, Connections, and Waiting Time.

**Table 7 Comparison between the proposed GTTRM model and previous works.**

| Feature | GTTRM | MOOP | MCTOPMTW | GATRS | PSO |
|---|---|---|---|---|---|
| Group split (Subgroups) | Dynamic | No | Static only | No | No |
| Time-centric personalization | Yes | No | Partial | No | No |
| Multi-value optimization | Yes | No | No | Partial | Partial |
| Waiting time consideration | Yes | No | Partial | No | No |
| Item constraints (ICDM) | Yes | No | No | No | No |

This broader set of factors makes the proposed GTTRM uniquely equipped to handle the complexities of modern travel planning, unlike MOOP, which considers only multi-value POIs without addressing connections or waiting time.

The comparison between the proposed GTTRM model and previous works is summarized in Table 7. It can be observed that the proposed model differs from the existing models not only in its handling of multiple values for POIs but also in its inclusion of transitions between POIs and waiting time considerations. In contrast:

- *GTTRM* provides a comprehensive approach by integrating multi-value POIs, multi-value connections, and waiting time, offering a more comprehensive solution for travel planning.
- *MOOP* lacks personalization and does not consider waiting time or multi-value transitions, focusing only on maximizing benefits within POI categories.
- *MCTOPMTW* supports multiple constraints but lacks an effective aggregation mechanism like GTTRM's traveler satisfaction metric, making it more complex and less intuitive.

Although existing models such as MOOP and MCTOPMTW have certain strengths, they are limited in scope. Our proposed model provides a more comprehensive and personalized approach by incorporating multi-value POIs, connections, and waiting time, making it better suited to meet the needs of travelers in practical, real-world scenarios.

The comparative analysis is further expanded to include additional models, such as the Genetic Algorithm-based Tour Recommendation System (GATRS) and the Particle Swarm Optimization-based approach (PSO). Table 7 demonstrates the superiority of the GTTRM in terms of personalization and the consideration of multiple factors such as time and waiting times, which are not addressed by other models like GATRS and PSO.

### Complexity analysis

The time complexity of the proposed GTTRM is derived from the complexity of the ACO algorithm. Given that the ACO iteratively optimizes the path selection based on pheromone updates, the time complexity of ACO is $O(n \log n)$, where $n$ is the number of POIs to be visited. This makes the proposed GTTRM more efficient than traditional models, such as the MCMTOPTW, which has a time complexity of $O(n^3)$ due to its reliance on exhaustive search algorithms. To further clarify the computational efficiency of GTTRM, we compared the complexity of our model with other relevant schemes. As

**Table 8 Comparison of time complexity between the proposed GTTRM model and relevant models.**

| Model | Time complexity (Big-O) |
| --- | --- |
| MCMTOPTW | $O(n^3)$ |
| *Anagnostopoulos et al. (2017)* | $O(n^2)$ |
| Proposed GTTM | $O(n \log n)$ |

**Table 9 The description of group members.**

| Number | Group members | | Gender number | | Relationship |
| --- | --- | --- | --- | --- | --- |
| | Adult number | Children number | M | F | |
| 1 | 2 | 3 | 2 | 3 | Family |
| 2 | 6 | 0 | 4 | 2 | Colleagues |
| 3 | 2 | 0 | 1 | 1 | Young couple |
| 4 | 4 | 0 | 2 | 2 | Retired friends |
| 5 | 15 | 0 | 6 | 9 | Students |

**Table 10 The relationship value in the first group.**

| | $U_1$ | $U_2$ | $U_3$ | $U_4$ | $U_5$ |
| --- | --- | --- | --- | --- | --- |
| $U_1$ | 1 | 1.5 | 2 | 2 | 2 |
| $U_2$ | 1.5 | 1 | 2 | 2 | 2 |
| $U_3$ | 2 | 2 | 1 | 2 | 2 |
| $U_4$ | 2 | 2 | 2 | 1 | 2 |
| $U_5$ | 2 | 2 | 2 | 2 | 1 |

shown in Table 8, GTTRM achieves a balance between computational efficiency and flexibility, making it a more practical solution for large-scale group travel recommendations.

## EXPERIMENTAL RESULTS

In this study, two experiments are conducted employing aggregation methods. The first experiment utilized the Average aggregation method to amalgamate all group members into a single profile. On the other hand, the second experiment applied the GACO algorithm (refer to "The Proposed Algorithm").

### Benchmark instances

Due to the absence of an existing dataset for GRSs in travel domains, a real-world dataset is compiled. Such dataset, designated as Durham, UK, indicates the geographical location of data collection.

Table 9 provides a detailed overview of the different group scenarios and sizes used in the experiments. Additionally, Tables 10–14 present the social relationship values

**Table 11 The relationship value in the second group.**

|  | $U_1$ | $U_2$ | $U_3$ | $U_4$ | $U_5$ | $U_6$ |
|---|---|---|---|---|---|---|
| $U_1$ | 1.0 | 1.2 | 1.2 | 1.8 | 1.1 | 1.2 |
| $U_2$ | 2.0 | 1.0 | 1.6 | 1.5 | 1.5 | 1.4 |
| $U_3$ | 1.7 | 1.5 | 1.0 | 1.4 | 1.8 | 1.6 |
| $U_4$ | 1.9 | 1.6 | 1.2 | 1.0 | 1.6 | 1.2 |
| $U_5$ | 1.6 | 1.9 | 2.0 | 1.9 | 1.0 | 1.4 |
| $U_6$ | 1.6 | 1.3 | 1.5 | 1.8 | 1.6 | 1.0 |

**Table 12 The relationship value in the third group.**

|  | $U_1$ | $U_2$ |
|---|---|---|
| $U_1$ | 1.0 | 2.0 |
| $U_2$ | 2.0 | 1.0 |

**Table 13 The relationship value in the fourth group.**

|  | $U_1$ | $U_2$ | $U_3$ | $U_4$ |
|---|---|---|---|---|
| $U_1$ | 1.0 | 2.0 | 2.0 | 1.0 |
| $U_2$ | 1.8 | 1.0 | 1.4 | 1.9 |
| $U_3$ | 1.4 | 1.0 | 1.0 | 1.3 |
| $U_4$ | 1.6 | 1.4 | 1.5 | 1.0 |

**Table 14 The relationship value in the fifth group.**

|  | $U_1$ | $U_2$ | $U_3$ | $U_4$ | $U_5$ | $U_6$ | $U_7$ | $U_8$ | $U_9$ | $U_{10}$ | $U_{11}$ | $U_{12}$ | $U_{13}$ | $U_{14}$ | $U_{15}$ |
|---|---|---|---|---|---|---|---|---|---|---|---|---|---|---|---|
| $U_1$ | 1.0 | 1.4 | 1.1 | 1.3 | 1.5 | 2.0 | 1.1 | 1.1 | 1.8 | 1.7 | 1.5 | 1.1 | 1.8 | 2.0 | 1.1 |
| $U_2$ | 1.3 | 1.0 | 1.9 | 1.7 | 1.3 | 1.9 | 1.9 | 2.0 | 1.3 | 1.4 | 1.4 | 1.2 | 1.3 | 1.2 | 1.8 |
| $U_3$ | 1.6 | 1.7 | 1.0 | 1.1 | 1.5 | 2.0 | 1.4 | 1.9 | 1.8 | 1.5 | 1.0 | 1.2 | 1.1 | 1.8 | 1.3 |
| $U_4$ | 1.9 | 1.7 | 1.0 | 1.0 | 1.8 | 1.2 | 2.0 | 1.9 | 1.6 | 1.1 | 1.0 | 1.8 | 1.3 | 1.8 | 1.6 |
| $U_5$ | 1.9 | 1.3 | 2.0 | 1.7 | 1.0 | 1.3 | 1.5 | 1.2 | 1.2 | 1.5 | 1.3 | 1.7 | 1.3 | 1.2 | 1.6 |
| $U_6$ | 1.3 | 1.9 | 1.0 | 1.6 | 1.3 | 1.0 | 1.6 | 1.7 | 1.4 | 1.7 | 1.0 | 1.8 | 1.7 | 1.2 | 1.5 |
| $U_7$ | 1.1 | 1.1 | 1.8 | 1.0 | 1.5 | 1.7 | 1.0 | 1.9 | 1.6 | 1.8 | 1.2 | 2.0 | 1.9 | 1.8 | 1.7 |
| $U_8$ | 1.7 | 1.0 | 1.4 | 1.7 | 1.6 | 1.4 | 1.4 | 1.0 | 1.4 | 1.7 | 1.6 | 1.9 | 1.3 | 1.6 | 1.1 |
| $U_9$ | 2.0 | 1.0 | 1.8 | 1.5 | 1.8 | 1.5 | 1.1 | 1.1 | 1.0 | 1.4 | 1.1 | 1.8 | 1.6 | 2.0 | 1.5 |
| $U_{10}$ | 1.6 | 1.7 | 1.9 | 1.3 | 1.4 | 1.1 | 1.4 | 1.2 | 1.4 | 1.0 | 1.4 | 1.5 | 1.2 | 1.3 | 1.5 |
| $U_{11}$ | 1.9 | 1.3 | 1.0 | 1.9 | 2.0 | 1.5 | 1.6 | 1.8 | 1.5 | 1.7 | 1.0 | 1.5 | 1.8 | 1.4 | 1.5 |
| $U_{12}$ | 1.4 | 1.4 | 1.9 | 1.1 | 1.8 | 1.7 | 1.9 | 1.1 | 1.2 | 1.2 | 1.2 | 1.0 | 1.5 | 1.2 | 1.9 |
| $U_{13}$ | 1.6 | 1.6 | 1.8 | 1.9 | 1.4 | 1.8 | 1.1 | 1.5 | 1.3 | 1.2 | 1.2 | 1.3 | 1.0 | 1.9 | 1.7 |
| $U_{14}$ | 1.7 | 1.9 | 2.0 | 2.0 | 1.8 | 1.7 | 1.8 | 1.2 | 1.6 | 1.1 | 1.5 | 1.9 | 1.1 | 1.0 | 1.8 |
| $U_{15}$ | 1.9 | 1.9 | 1.2 | 1.3 | 1.9 | 1.4 | 1.9 | 1.4 | 1.7 | 1.7 | 1.3 | 1.4 | 1.3 | 1.9 | 1.0 |

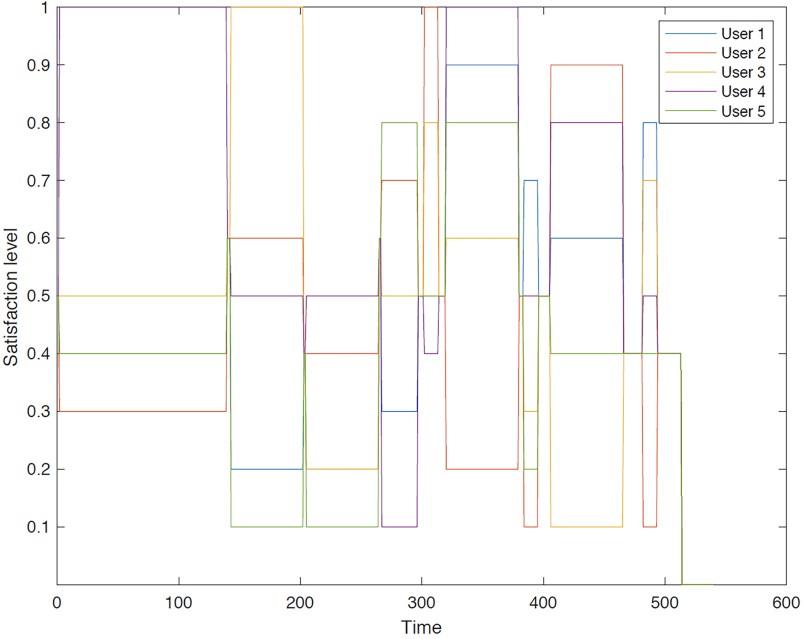

**Figure 3** Illustration of group 1's satisfaction levels under group aggregation.

among group members, which were randomly assigned within a range of 1 to 2. A value of 1 indicates the weakest relationship, while a value of two signifies the strongest relationship.

To ensure the robustness of our findings, we have tested the model extensively across a range of initial values and parameter settings. The GTTRM model was evaluated over multiple scenarios with varying relationship values, preferences, and subgroup configurations. Each test scenario consistently produced satisfactory results, indicating that the GTTRM is stable and effective under different initial conditions. Moreover, the ACO algorithm used in GTTRM inherently contributes to the model's robustness. ACO's iterative optimization process, which includes pheromone updating and path selection based on cumulative runs, ensures convergence to an optimal solution regardless of initial conditions. This multi-scenario approach has confirmed that the results are reliable, and no significant variations were observed across different runs, underscoring the stability and reliability of GTTRM's performance in real-world applications.

## Comparative analysis

This section provides a comparative analysis of two aggregation methods, revealing that the GACO algorithm surpasses other methods in terms of maximizing the satisfaction levels of group members.

### Group aggregation

The outcomes of the experiments on group aggregation is presented in this section, wherein group travelers are consolidated into a single profile. The satisfaction level for each traveler within the groups has been illustrated. Figures 3–7 depict the satisfaction levels for

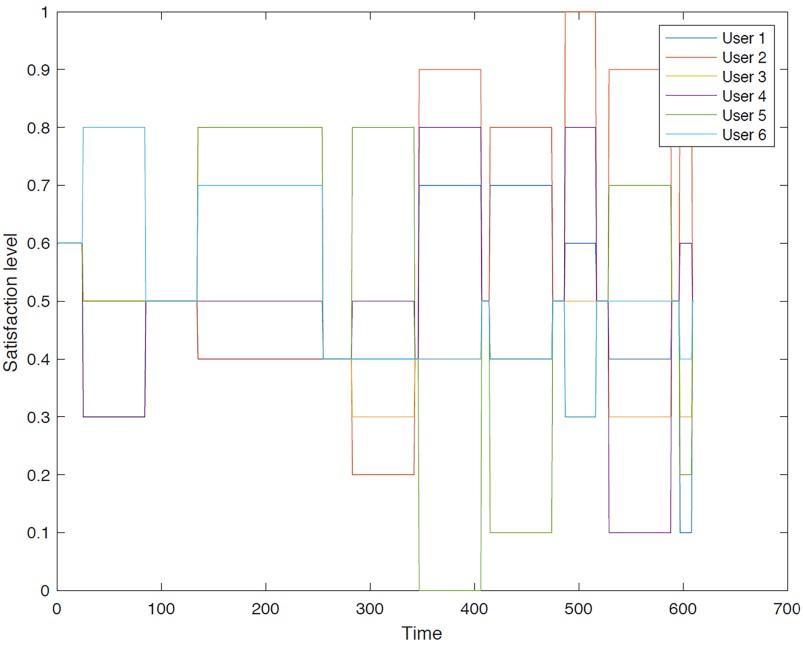

**Figure 4 Illustration of group 2's satisfaction levels under group aggregation.**

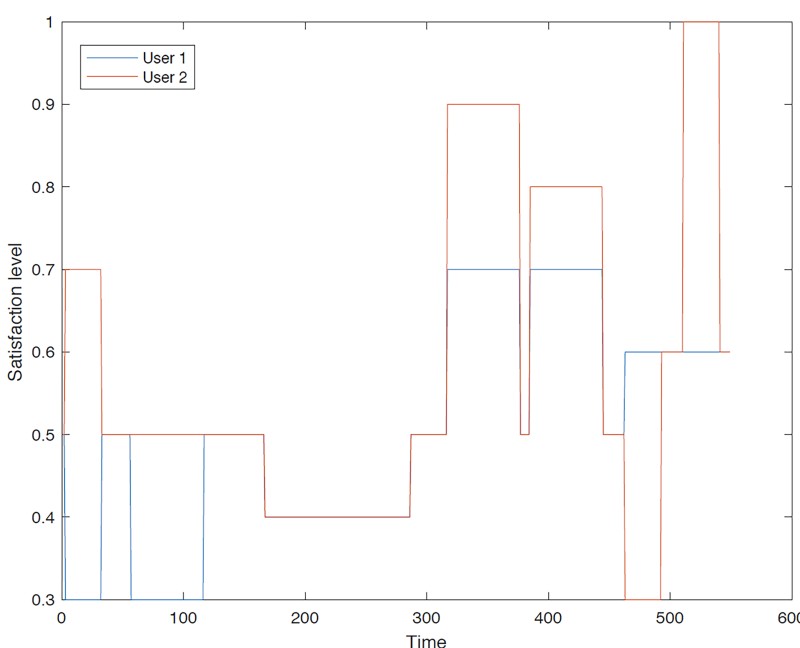

**Figure 5 Illustration of group 3's satisfaction levels under group aggregation.**

each traveler from the commencement to the conclusion of the tour, demonstrating the variability in satisfaction among group members.

Figure 3 visually represents the changes in satisfaction levels for all members of the first group over time. Each line corresponds to an individual user's satisfaction level,
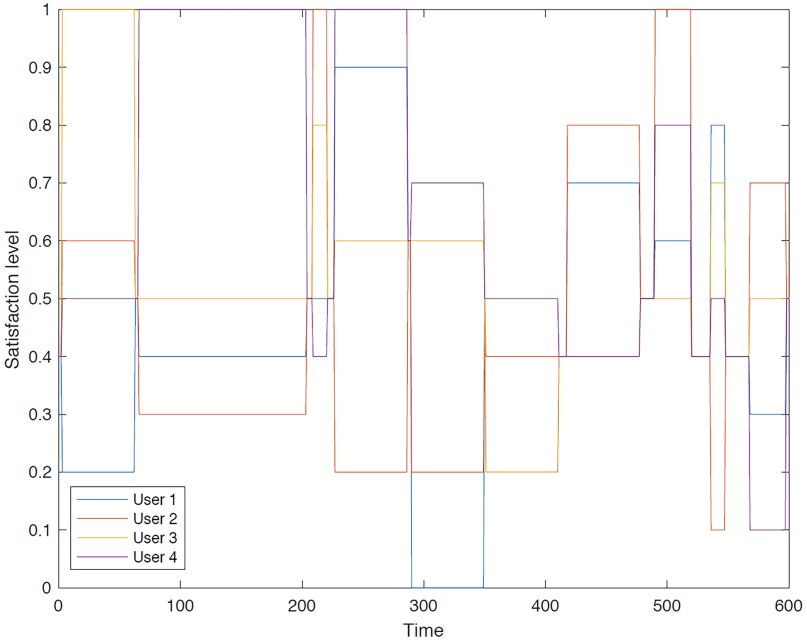

**Figure 6 Illustration of group 4's satisfaction levels under group aggregation.**

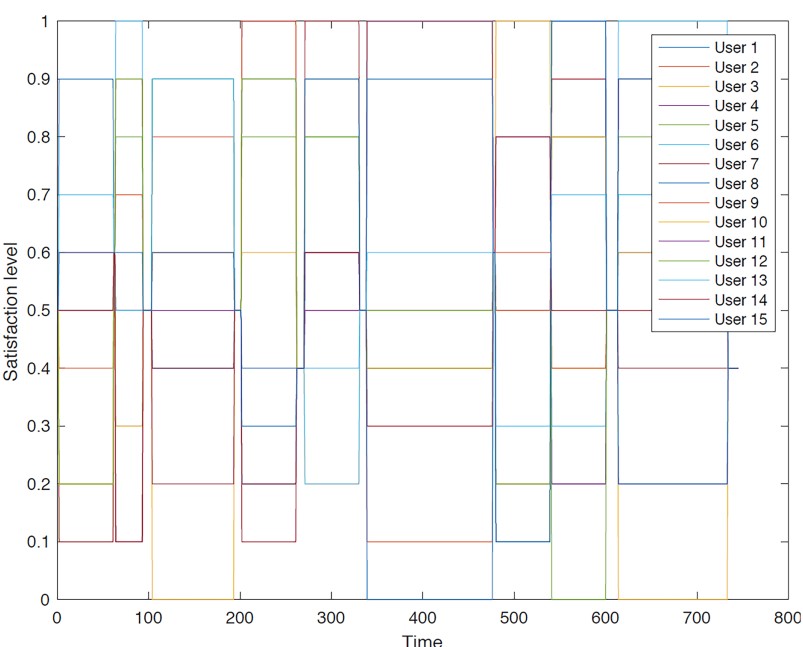

**Figure 7 Illustration of group 5's satisfaction levels under group aggregation.**

normalized to a maximum of 1. The varying satisfaction levels among users highlight the diverse preferences and constraints within the group, demonstrating the challenges of achieving optimal satisfaction for all members.

Figure 4 visually represents the overall satisfaction levels for all members of the second group. The y-axis represents the satisfaction level, while the x-axis indicates the group members. The data points and error bars illustrate the distribution of satisfaction levels within the group. The figure suggests that the majority of travelers do not achieve the highest satisfaction level under the group aggregation approach, as this approach consolidates all users' values into a single value, potentially leading to dissatisfaction for some members.

Figure 5 visually illustrates the satisfaction levels of traveler 1 and traveler 2 within the third group. The variation in satisfaction among the travelers is depicted from the commencement of the tour up to 150 min (with the tour duration segmented into minutes). It is evident that the satisfaction levels for both users in the final two-thirds of the trip duration are significantly higher compared to the initial third.

Figure 6 illustrates the diverse preferences of travelers within group 4, highlighting a scenario in which the group aggregation approach yields a trip which does not meet the satisfaction of all members.

Figure 7 illustrates the significant disparity in users' preferences due to the necessity of remaining together throughout the trip, as their distinct preferences are not taken into account.

### User aggregation

The outcomes of implementing the proposed GACO optimization algorithm within the proposed GTTRM are presented in this section.

**First group**: A comprehensive evaluation and comparison of satisfaction levels for each user is conducted under both user aggregation and group aggregation. The results consistently demonstrate that user aggregation, as determined by the GACO algorithm, generally outperforms group aggregation in terms of maximizing overall satisfaction. Figures 8–12 provide a detailed visualization of the satisfaction levels for members of the first group.

Figure 8 illustrates the satisfaction level of traveler 1 within the first group across different methodologies. It is evident that the user aggregation approach yields a higher level of satisfaction in comparison to the group aggregation method. Furthermore, the figure indicates the differential performance of these models; the group aggregation method bases its decisions on the group profile, whereas the user aggregation method derives its decisions from individual user profiles.

Figure 9 illustrates the satisfaction level of traveler 2 in the first group, where the traveler's satisfaction surpasses that of the group aggregation method. Although user aggregation demonstrates superior performance overall, there are instances where the group aggregation method yields better results over short periods. This occasional advantage of the group aggregation method can be attributed to the GACO algorithm's design, which sequentially selects individual POIs without considering the entire route collectively.

Figure 10 illustrates the satisfaction level of user 3 in the initial group, in which both methods yield comparable results. It also demonstrates that, at certain instances, the group

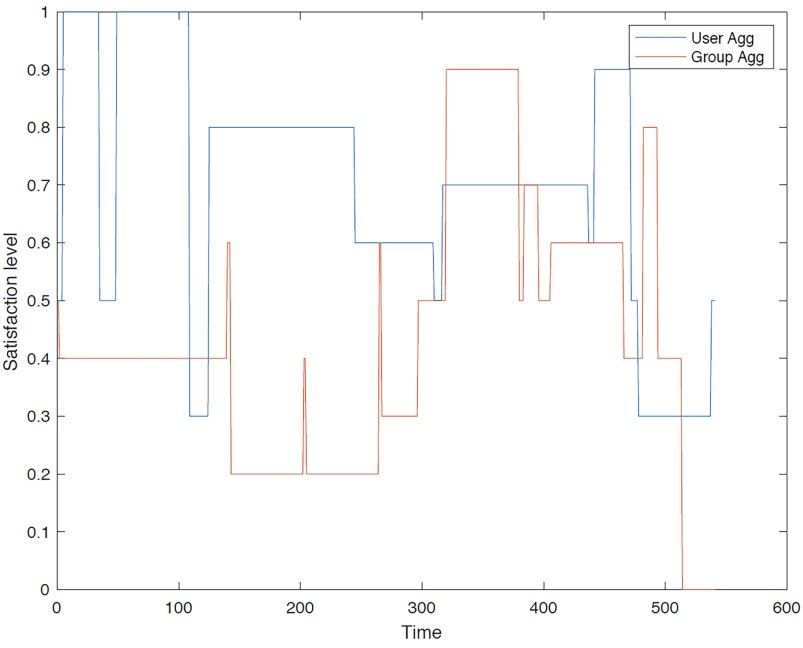

**Figure 8 Satisfaction levels for traveler 1 from group 1 under group and user aggregations.**

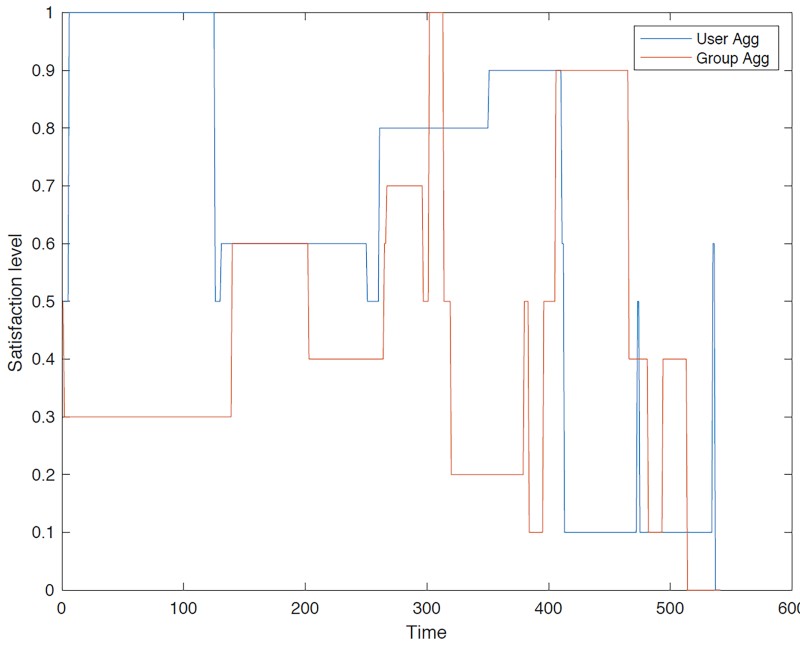

**Figure 9 Satisfaction levels for traveler 2 from group 1 under group and user aggregations.**

aggregation method outperforms the user aggregation method. This occurs because the user aggregation algorithms may select a POI that subsequently limits the availability of other suitable POIs, resulting in the selection of a POI with lower satisfaction.

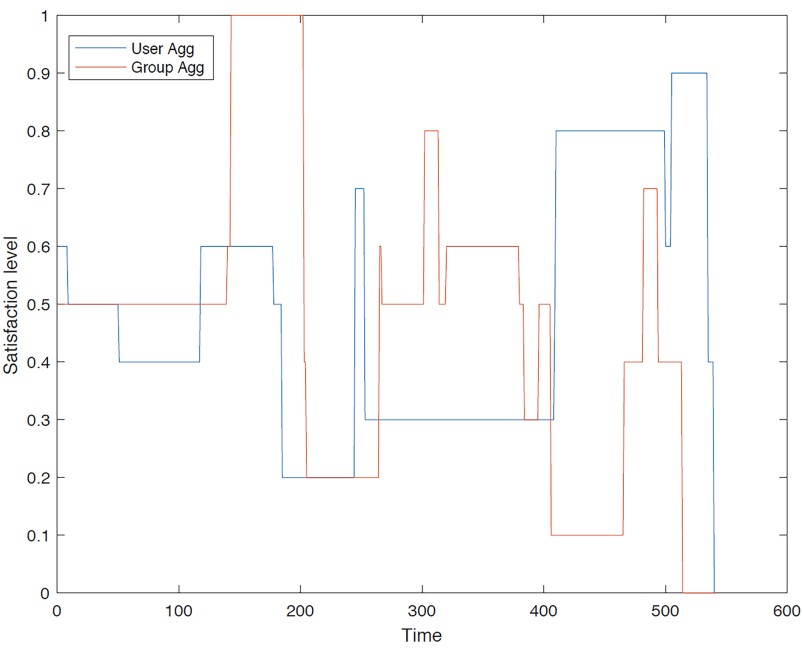

**Figure 10 Satisfaction levels for traveler 3 from group 1 under group and user aggregations.**

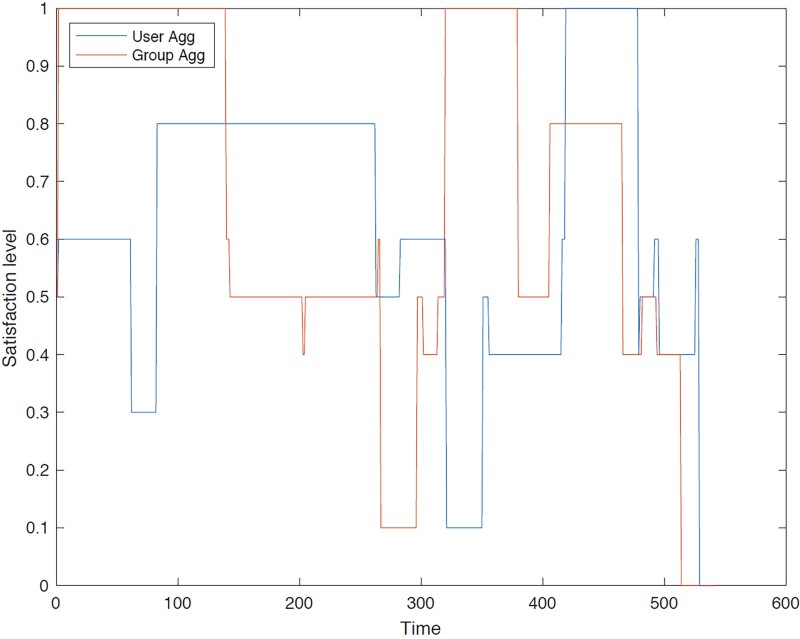

**Figure 11 Satisfaction levels for traveler 4 from group 1 under group and user aggregations.**

Figure 11 illustrates a comparison of the two methods concerning traveler 4's satisfaction levels, indicating similar outcomes for both methods. Initially, the group method outperforms the user method, primarily because traveler 4 has a

strong relationship with a group member selected by the proposed GACO algorithm.

Figure 12 illustrates the satisfaction level for traveler 5 within group 1, demonstrating that the user aggregation approach sometimes yields superior results, while at other times, the group aggregation method proves more effective. As previously discussed in the context of User 3, the GACO algorithm may select a POI in an area of the city where surrounding POIs offer low satisfaction.

**Second group**: In the second instance, we evaluated the satisfaction levels for the members of group 2 using group aggregation method. Figures 13–18 provide a comparative analysis between user aggregation and group aggregation based on satisfaction levels.

Figure 13 illustrates the distinctions between group aggregation and user aggregation. The latter yields superior results, whereas the former results in an extended tour. It is evident that user aggregation concludes the tour sooner than group aggregation method. Overall, user aggregation generally provides a higher level of satisfaction for the trip.

Figure 14 illustrates a comparison of the outcomes generated by the different aggregation approaches. The user aggregation approach yields superior overall results, whereas the group aggregation method results in an extended tour. The GACO algorithm is constrained to select sequences of POIs that are visited simultaneously, thereby ensuring a consistent level of satisfaction. The primary reason for this outcome is that GACO selects POIs on an individual basis.

Figure 15 illustrates that the user aggregation approach yields a recommended trip of shorter duration compared to the tour derived from group aggregation. As depicted in the figure, group aggregation offers a more favorable outcome for User 3. The primary reason for this is that the GACO algorithm selected User 3 to pair with another user based on their high relationship value, disregarding User 3's individual preferences.

Figure 16 illustrates that the user and group aggregation approaches yield comparable tours in view of overall satisfaction. The GACO algorithm selected traveler 4 to accompany another traveler in the group for a segment of the tour, resulting in a slightly lower satisfaction level compared to the overall group aggregation.

Moreover, Fig. 17 demonstrates that both group and user aggregation approaches result in a high satisfaction for traveler 5 within the second group. When the proposed algorithm selects a POI in a particular direction, the subsequent point may not meet traveler's satisfaction due to the limited number of options available in that direction. It is observed that the initial segment of the trip, as determined by user aggregation, exhibits a high satisfaction level; however, GACO struggles to identify another suitable POI to maintain this level of satisfaction consistently.

Figure 18 reveals that the group aggregation method initially outperforms the user aggregation approach at the beginning of the tour. However, as the trip progresses, the user aggregation approach demonstrates superior performance, particularly at the midpoint. This can be attributed to the fact that user 6 is paired with another user within the same group, leading to potential conflicts or misalignments in preferences that can impact satisfaction levels under the user aggregation approach.

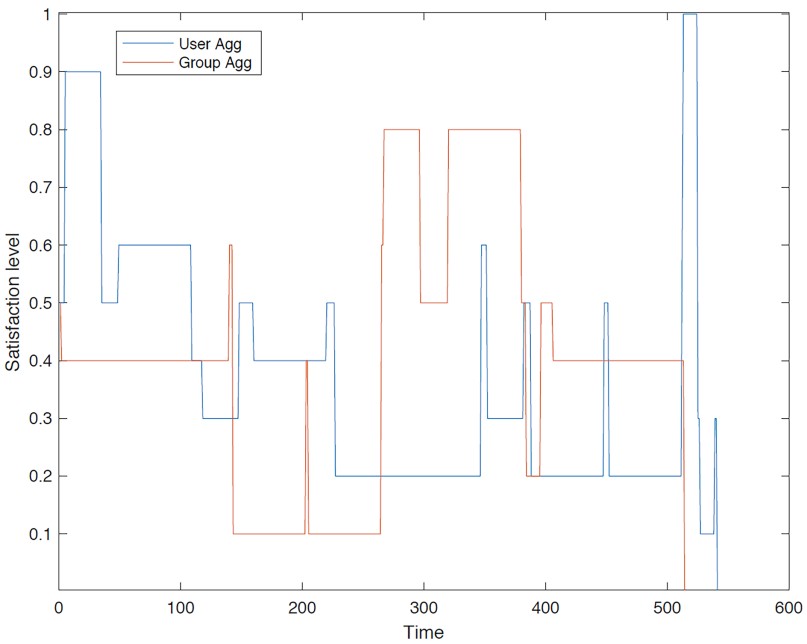

**Figure 12** Satisfaction levels for traveler 5 from group 1 under group and user aggregations.

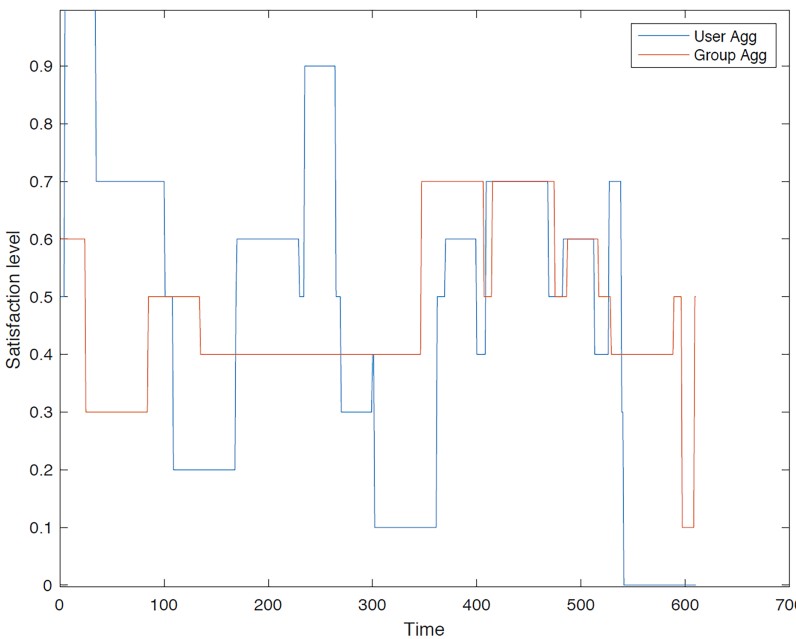

**Figure 13** Satisfaction levels for traveler 1 from group 2 under group and user aggregations.

## Discussion

In this article, multiple experiments are conducted on the two aggregation methods. The results from the user aggregation method surpass those from the group aggregation method. For instance, Fig. 8 illustrates the comparative performance of these aggregation

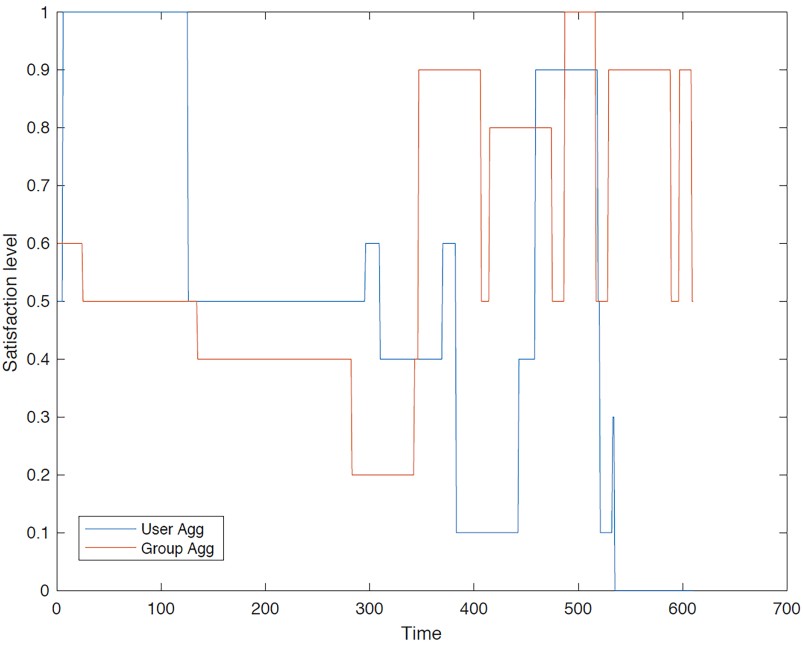

**Figure 14 Satisfaction levels for traveler 2 from group 2 under group and user aggregations.**

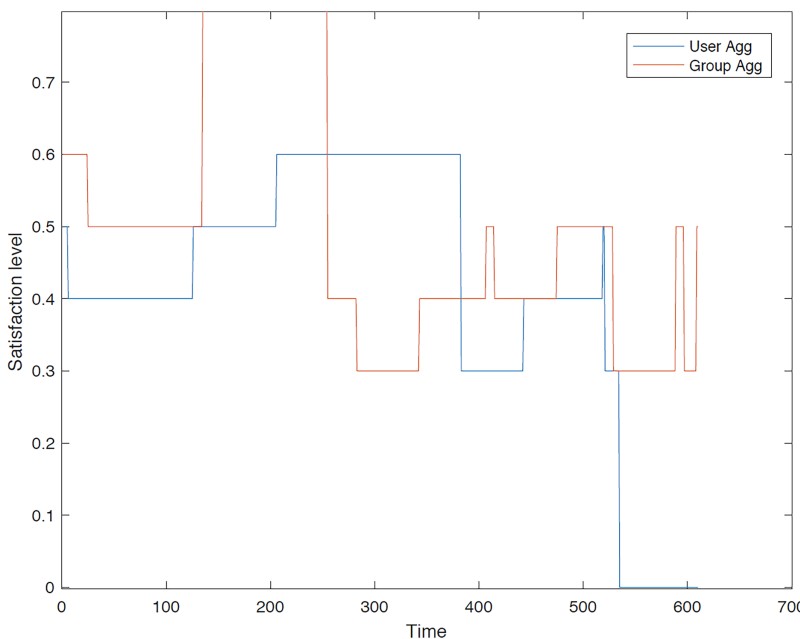

**Figure 15 Satisfaction levels for traveler 3 from group 2 under group and user aggregations.**

methods, highlighting that the user aggregation method excels in maximizing the traveler's satisfaction level. Additionally, the first and second traveler within the groups consistently exhibited higher level of satisfaction as compared to other group members. The proposed GTTRM was validated under diverse test conditions to ensure that it performs

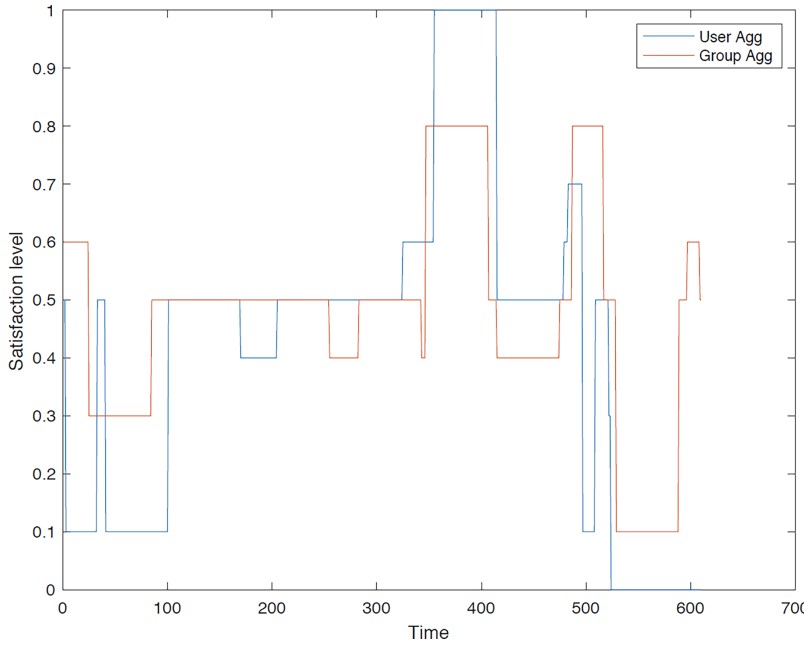

**Figure 16 Satisfaction levels for traveler 4 from group 2 under group and user aggregations.**

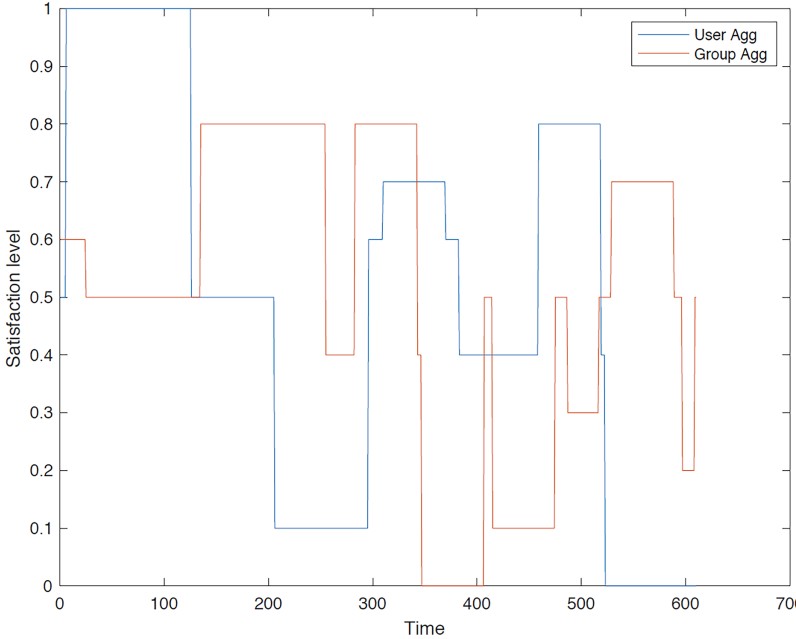

**Figure 17 Satisfaction levels for traveler 5 from group 2 under group and user aggregations.**

consistently. Through repeated scenario testing with various initial values, we confirmed that GTTRM reliably maximizes group satisfaction and dynamically adapts to subgroup configurations. These results are not subject to random variation due to single-run dependencies; rather, the consistent results observed across scenarios demonstrate the

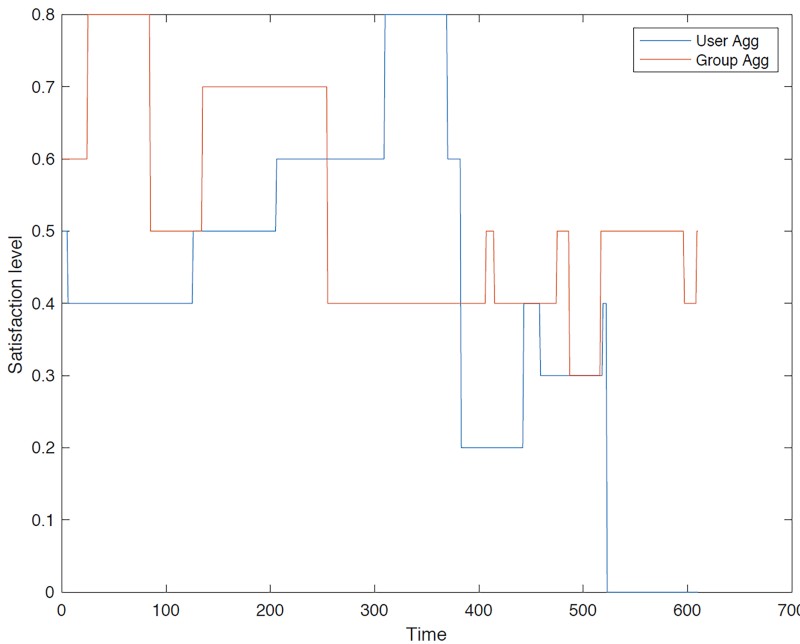

**Figure 18 Satisfaction levels for traveler 6 from group 2 under group and user aggregations.**

robustness of our model. Future work may involve further statistical testing, but the current approach provides a stable and effective foundation for group travel recommendations. In summary, our findings demonstrate that the GACO algorithm achieves superior outcomes (*i.e.*, higher level of satisfaction for travelers) relative to the group aggregation approach, particularly for the first and second traveler in the group. This enhanced performance can be attributed to the social relationship values integrated within the algorithm. These values drive the algorithm to prioritize the satisfaction of the first traveler who is accompanied by a second traveler, who may not derive full satisfaction from visiting the POIs preferred by the first user.

### Limitations of the GTTRM

While the proposed GTTRM offers significant improvements in handling group travel recommendations and dynamic subgroup formation, there are certain limitations that should be considered. One limitation is the dependency on accurate preference data for effective subgroup formation. If group members do not provide sufficient or accurate preference data prior to the trip, the model may struggle to form meaningful subgroups, leading to suboptimal satisfaction outcomes. Another potential limitation is the frequency of preference changes. The model assumes that subgroup formation can adapt to evolving preferences throughout the trip. However, if preferences change too frequently, the algorithm may require excessive recalculations, leading to increased computational overhead and potentially reducing the system's real-time performance. In addition, the model may not perform as effectively in groups where interpersonal relationships heavily influence decisions. The current implementation of ACO focuses on optimizing

satisfaction based on preferences, but does not explicitly account for social dynamics or group decision-making hierarchies, which could affect the outcome in real-world group travel scenarios.

## CONCLUSION

This study introduced the group tour trip recommender model, a novel solution to the group tour trip design problem, with a particular focus on subgroup formation and conflict resolution. The use of group ant colony optimization allows for flexible and efficient optimization of group satisfaction while accommodating individual constraints. Comparative analysis with existing models demonstrates that GTTRM outperforms in both satisfaction levels and computational efficiency. Future research will focus on integrating real-time user feedback and expanding the model to support larger, more diverse groups.

### Funding

This study is supported *via* funding from Prince Sattam bin Abdulaziz University project number (PSAU/2023/R/1445). The funders had no role in study design, data collection and analysis, decision to publish, or preparation of the manuscript.

### Grant Disclosures

The following grant information was disclosed by the authors:
Prince Sattam bin Abdulaziz University: PSAU/2023/R/1445.

### Competing Interests

The authors declare that they have no competing interests.

### Author Contributions

- Mohammed Alatiyyah conceived and designed the experiments, performed the experiments, analyzed the data, performed the computation work, prepared figures and/ or tables, authored or reviewed drafts of the article, and approved the final draft.

### Data Availability

  The code is available in the Supplemental Files.

### Supplemental Information

Supplemental information for this article can be found online at http://dx.doi.org/10.7717/ peerj-cs.2589#supplemental-information.

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
