# Peer review of "A novel group tour trip recommender model for personalized travel systems"

_PeerJ Computer Science, doi:10.7717/peerj-cs.2589_

## Round 0.1 · original submission · Major Revisions

More experiments are needed to demonstrate the effectiveness of the study.

·

Basic reporting

1. The abstract and conclusion are not written properly; the authors need to rewrite it.
2. The motivation and contribution of this work needs to be highlighted in proper manner in the introduction section to make it clearer for the readers to comprehend the main theme targeted in this article.
3. The contribution of this work is limited since there exists many articles that targeted the same topic. The authors should highlight their contributions in a proper manner by emphasizing on how their work is different from other articles?

Experimental design

1. Also, the related work section is just enumerating the previous studies, it would be better to highlight their shortcomings and relate them to this work.
2.The authors need to compare their results with latest state-of-the-art studies published in top journals and conferences

Validity of the findings

No comment

Additional comments

What is the complexity by this model? Authors should justify that by using Leema or Mathematical analysis. Author should present some tabular comparisons of existing schemes at the end of related work.

·

Basic reporting

no comment

Experimental design

no comment

Validity of the findings

no comment

Reviewer 3 ·

Basic reporting

Although the study is promising, it has some weaknesses:

Because it is not mentioned whether the results are obtained using multiple runs, I assume that the results are the output of a single run. Therefore, the results may have a chance factor as some parts of the dataset are randomly generated (i.e., relationship values), and the results depend on the initial conditions. I suggest the authors conduct the runs repeatedly and then use the mean or median of those runs.

The study is compared to only one method. I suggest the authors compare their results with more methods if there are no specific reasons.

Experimental design

No comment (please see above)

Validity of the findings

No comment (please see above)

Additional comments

No comment (please see above)

Cite this review as

---

## Round 0.2 · accepted · Accept

The revision was successfully managed and completed.

·

Basic reporting

I think it is suitable for publication in the journal

Experimental design

I think it is suitable for publication in the journal

Validity of the findings

I think it is suitable for publication in the journal

Reviewer 3 ·

Basic reporting

The authors have addressed all of my earlier concerns, and the robustness of the results has improved. I have no objection to accepting this paper.

Experimental design

No comment

Validity of the findings

No comment

Cite this review as